# Genome-Wide Detection and Analysis of Copy Number Variation in Anhui Indigenous and Western Commercial Pig Breeds Using Porcine 80K SNP BeadChip

**DOI:** 10.3390/genes14030654

**Published:** 2023-03-05

**Authors:** Chengliang Xu, Wei Zhang, Yao Jiang, Mei Zhou, Linqing Liu, Shiguang Su, Xueting Li, Chonglong Wang

**Affiliations:** 1Key Laboratory of Pig Molecular Quantitative Genetics of Anhui Academy of Agricultural Sciences, Anhui Provincial Key Laboratory of Livestock and Poultry Product Safety Engineering, Institute of Animal Husbandry and Veterinary Medicine, Anhui Academy of Agricultural Sciences, Hefei 230031, China; 2Faculty of Animal Science and Technology, Yunnan Agricultural University, Kunming 650500, China; 3National Animal Husbandry Service, Beijing 100125, China

**Keywords:** copy number variation, Anhui indigenous pig breeds, Western commercial pig breeds, SNP, immune, meat quality

## Abstract

Copy number variation (CNV) is an important class of genetic variations widely associated with the porcine genome, but little is known about the characteristics of CNVs in foreign and indigenous pig breeds. We performed a genome-wide comparison of CNVs between Anhui indigenous pig (AHIP) and Western commercial pig (WECP) breeds based on data from the Porcine 80K SNP BeadChip. After analysis using the PennCNV software, we detected 3863 and 7546 CNVs in the AHIP and WECP populations, respectively. We obtained 225 (loss: 178, gain: 47) and 379 (loss: 293, gain: 86) copy number variation regions (CNVRs) randomly distributed across the autosomes of the AHIP and WECP populations, accounting for 10.90% and 22.57% of the porcine autosomal genome, respectively. Functional enrichment analysis of genes in the CNVRs identified genes related to immunity (*FOXJ1*, *FOXK2*, *MBL2*, *TNFRSF4*, *SIRT1*, *NCF1*) and meat quality (*DGAT1*, *NT5E*) in the WECP population; these genes were a loss event in the WECP population. This study provides important information on CNV differences between foreign and indigenous pig breeds, making it possible to provide a reference for future improvement of these breeds and their production performance.

## 1. Introduction

With the development of DNA sequencing technology and other molecular biology techniques, different types of genetic variations can be investigated at the DNA level such as single nucleotide polymorphism (SNP), insertion-deletion (indel), microsatellite DNA, copy number variations (CNVs), and structural variations in large indels [1]. These rich variants are the driving force of biological evolution and one of the factors that cause genetic and phenotypic diversity. CNVs are mutations of DNA fragments ranging from kilobases to megabases in length, which are prevalent in human and domestic animal genomes [2]. In recent years, most studies have considered SNPs as the main genetic variants contributing to phenotypic diversity [3], while structural variations have scarcely been investigated despite their significant impact on explaining phenotypic variation. Speciation is correlated with reduced gene flow among genetically differentiated populations. Paudel et al. [4] identified 1408 CNV regions (CNVRs) in *Sus domesticus* and determined that CNVs evolve approximately 2.5 times faster than SNPs. Therefore, studying CNVs in the genome is useful for understanding the genetic diversity of genomes, the genomic makeup of different species, and individual differences.

In previous studies, it was found that CNVs mainly occurred during meiosis. They were caused by chromosomal structural rearrangements resulting in non-allelic homologous recombination that could be passed to offspring [5], with some CNVs occurring during the period of DNA self-repair. Lee et al. proposed the fork stalling and template switching model to explain some of the complex CNV formation mechanisms [6]. CNVs have now been identified in approximately 12% and 7% of the human [7] and mouse genomes [8], respectively, and studies have successfully detected a large number of CNVs in domestic animals. Many studies have reported that CNVs affect protein-coding genes in pigs [9], humans [10], mice [11], cattle [12], and other species.

Because CNVs cover a broader genomic region in domestic animals, many studies have reported that CNVs play an important role by potentially altering gene structure and dosage, exposing recessive alleles, alternating gene regulation, and affecting other mechanisms [13]. For example, a large segment of duplication on the Z chromosome of chickens leads to blocked chicken feather development and late feathering [14]. A porcine study identified a significant relationship between the copy number gain of the proto-oncogene (c-KIT) on chromosome 8 [15]. In addition, many studies have found that some CNVs have a significant impact on various complex characteristics and biological processes, including phenotypic and important economic traits, such as milk production [16], reproductive [17,18], and growth traits [19]. We, therefore, consider it important to use CNV studies for animal genetic improvement.

Western commercial pig (WECP) breeds have advantages in growth rate, daily gain, and other features, whereas Anhui indigenous pig (AHIP) breeds have advantages in meat quality, disease resistance, and other traits. Our group previously evaluated runs of homozygosity and SNP signatures of selection, identifying many differential gene expression patterns between AHIP and WECP populations. Currently, there are no reports of CNVs between the AHIP and WECP populations, nor are there any reports of differences between the two populations caused by CNVs. This study used the Illumina Porcine 80K SNP BeadChip and Penn CNV software to identify autosomal CNVs in 320 pigs from the AHIP and WECP populations. We analyzed the distribution of CNVs in the pig genome and different populations to comprehensively determine the population genetic background differences and genetic diversity of AHIPs and WECPs, providing crucial information for the genetic improvement of these breeds through sustainable breeding programs.

## 2. Materials and Methods

### 2.1. Ethics Statement

All experiments in this study were carried out in accordance with the recommendations of the Animal Care Committee of the Anhui Academy of Agricultural Sciences (Hefei, China). The program was approved by the Animal Protection Committee of Anhui Provincial Agricultural Science (Hefei, China; No. AAAS2020-04).

### 2.2. Samples and Genotyping

A total of 320 blood samples were collected from 10 breeds of pigs, including 5 foreign breeds (WECP): Yorkshire (n = 50), Landrace (n = 30), Duroc (n = 30), Pietrain (n = 30), Berkshire (n = 30), and 5 indigenous Anhui pig populations (AHIP): Anqing six-end-white pig (ASP, n = 30), Wei pig (WP, n = 30), Wannan Spotted pig (WSP, n = 30), Wannan Black pig (WBP, n = 30), and Huai pig (HP, n = 30). We considered two different groups based on the results obtained in the previous study [20]. We extracted genomic DNA from ear tissues using the traditional phenol/chloroform method. The concentration and purity of genomic DNA were assessed using a NanoDrop™ 2000 (Thermo Fisher Scientific, Waltham, MA, USA).

The Porcine SNP80k BeadChip (Illumina, San Diego, CA, USA) includes 68,528 SNPs with an average gap length of 38 kb on each chromosome and was used for individual genotyping. GenomeStudio version 2011 (Illumina, version 1.9.4) was used for genotype calling and SNP clustering, and tight quality control was applied to SNP filtering to improve the precision of CNV detection. SNPs with call rates below 90%, minor allele frequencies below 0.03, and *p*-values below 1e6 for Hardy-Weinberg equilibrium were eliminated. SNPs found on the sex chromosomes were also disregarded. Ultimately, 54,075 autosomal SNPs were retained for CNV analysis and detection.

### 2.3. Genome-Wide Detection of CNVs and CNVRs

To discover CNVs, the population frequency of the B allele (PFB) of SNPs and the log R ratio (LRR) and B allele frequency (BAF) at each SNP marker were incorporated into a hidden Markov model using the PennCNV software [21]. Each SNP, LRR and BAF was exported from GenomeStudio, and the PFB was created based on each SNP BAF. Based on the Scrofa 10.2 reference genome assembly (http://may2015.archive.ensembl.org/index, accessed on 2 June 2022), the physical locations of SNPs on chromosomes were established. Using the gcmodel option in PennCNV, genomic waves were adjusted for the GC content in the 500 kb genomic region around each SNP on both sides. Subsequently, each sample was subjected to quality control before analysis to reduce the possibility of false-positive CNVs [22]. In summary, we included samples with LRR 0.3, BAF drift 0.01, and an LRR 0.05 for the GC wave factor. Copy number variations were classified as having at least two individuals and three or more consecutive SNPs [23].

The software CNVRuler was then used to integrate the overlapping CNVs to create CNVRs [24]. We chose the CNV method by region approach for our investigation. Subsequently, we applied a recurrence value of 0.3, as advised by the CNVRuler manual and previous studies [25], to avoid overestimating the size and frequency of CNVRs. Finally, control raw data filtered for occurrence in at least three individuals were used as the last of our results.

### 2.4. Gene Contents and Functional Annotation

We used BioMart (http://www.biomart.org/, accessed on 15 October 2022) to identify genes overlapping with the CNVRs, including gene ID and gene type from the Ensembl gene 80 database based on the Sscrofa 10.2 genome assembly. Owing to the incomplete annotation of porcine genes in the Gene Ontology (GO) or Kyoto Encyclopedia of Genes and Genomes (KEGG) databases, we first converted the pig Ensembl gene IDs to orthologous human Ensembl gene IDs using BioMart [26]. The Metascape tool (https://metascape.org/, accessed on 3 November 2022) was used for gene enrichment analysis, including GO analysis and KEGG pathway analysis (min overlap: 3, *p* value cutoff: 0.01, min enrichment: 1.5). In this study, we annotated gene sets according to the type of CNV, based on their formation mechanisms, and those with a corrected *p*-value of <0.05 were selected to explore the genes involved in biological processes. In addition, information from the pig quantitative trait loci (QTL) database was used to find QTL located in the identified CNVRs or QTL partially overlapping with the CNVRs (at least 50% inside the CNVs) for further analysis (https://www.animalgenome.org/cgi-bin/QTLdb/SS/index, accessed on 5 November 2022).

## 3. Results

### 3.1. Genome-Wide Detection of CNVs

Following a strict calling pipeline in PennCNV, we discovered 11,409 CNVs in 18 chromosomes, including 7546 CNVs in the WECP population (loss: 6644, gain: 902) and 3863 CNVs in the AHIP population (loss: 3368, gain: 495). A total of 604 CNVRs were discovered by merging the CNV areas that had been determined to overlap, including 379 and 225 events in the WECP and AHIP populations, respectively. The statistical mapping of CNVRs in autosomes of the AHIP and WECP populations is shown in Figure 1.

Among these 604 CNVRs, we found 133 gain and 471 loss events. The distribution of CNVRs in pig autosomes is shown in Table 1. The distribution of these CNVRs on chromosomes is shown in Figure 2 (Appendix A). We also found that in the WECP population, these CNVRs ranged in length from 4.85 kb to 9761.85 kb, with a median of 901.97 kb and a total coverage of 553.08 Mb, corresponding to 22.57% of the pig genome. The number of CNVRs distributed on chromosome 1 was large, with 39 CNV events. In the AHIP population, the CNVRs ranged in length from 25.44 kb to 6400.56 kb, with a median of 774.74 kb and a total coverage of 267.18 Mb, corresponding to 10.90% of the pig genome (Appendix A). The number of CNVRs distributed on chromosome 1 was greatest with 24 CNV events. The overall size distribution showed smaller CNVRs in the AHIP population than in the WECP population (Figure 3).

### 3.2. Analysis of Genes within AHIPs and WECPs CNVR

In the AHIP and WECP populations, we retrieved 720 and 1749 genes within or overlapping CNVRs, respectively, using the Ensembl genes 80 databases (reference genome: Sscrofa 10.2, Appendix A). To facilitate subsequent analyses, we converted these genes to Sscrofa 11.1 Duroc and ultimately obtained 408 and 1113 genes in the AHIP and WECP populations, respectively (Figure 4A) (Appendix A). The two populations had 277 common genes, with 80 unique genes in the AHIP populations and 639 unique genes in the WECP populations. Among them, AHIPs_loss, AHIPs_gain, WECPs_loss, and WECPs_gain contained 26, 86, 75, and 44 unique genes, respectively (Figure 4B).

### 3.3. Enrichment Analysis of Candidate Genes in WECPs and AHIPs

We performed enrichment analysis of genes in the WECP and AHIP populations based on the type of CNVR events. The results of GO enrichment analysis showed that these genes were mainly enriched in the regulation of vesicle-mediated transport (GO: 0060627, *p* = 2.69 × 10^−4^), regulation of secretion (GO: 0051046, *p* = 3.98 × 10^−4^), regulation of immune effector process (GO: 0002697, *p* = 4.58 × 10^−4^), and actin cytoskeleton organization (GO: 0030036, *p* = 9.40 × 10^−3^). The KEGG pathway analysis showed enrichment in inositol phosphate metabolism (hsa00562, *p* = 3.13 × 10^−2^), cAMP signaling pathway (hsa04024, *p* = 3.13 × 10^−2^), and pathways of neurodegeneration-multiple diseases (hsa05022, *p* = 3.13 × 10^−2^) that were loss events in the WECP populations (Figure 5A,B). GO enrichment analysis also showed genes that were mainly enriched in the regulation of leukocyte mediated cytotoxicity (GO: 0001910, *p* = 2.78 × 10^−2^), the regulation of centrosome duplication (GO: 0010824, *p* = 2.89 × 10^−2^), regulation of mitotic cell cycle (GO: 0007346, *p* = 2.89 × 10^−2^) and the sensory perception of umami taste (GO: 0050917, *p* = 4.55 × 10^−2^). These were identified as loss events in the AHIP populations (Figure 5C,D) (Appendix A).

## 4. Discussion

Pigs play an important role in human development and are the most important meat source in the world. China has a long history of artificial domestication and has the greatest resource of pig breeds in the world, contributing significantly to the improvement of other pig species around the world. In recent years, joint genome-wide association studies have found a large number of potential candidate markers associated with economic and phenotypic traits that are important for genetic improvement of breeds. Compared with SNPs, CNVs have unique advantages in that they have longer DNA segments and affect gene expression more easily; therefore, they can be applied as a new molecular genetic marker to the study of genetic mechanisms in animals [27]. In domestic animals, several studies have revealed CNVs that can affect phenotypic qualities, economic traits, and the prevalence of a number of serious diseases [28,29,30]. In our study, we performed a genome-wide detection analysis of AHIP and WECP populations based on the Porcine 80K SNP BeadChip. After the Penn CNV software analysis, we detected 11,409 CNVs on autosomes. We employed a significantly strict definition of overlap by CNVRuler software processing, which ultimately obtained 604 CNVRs. However, there were some CNVRs that appeared only at low frequency in the population, which we considered likely to be caused by the large number of breeds in each population. A second reason may be that the SNP BeadChip has inherent limitations in resolution and sensitivity when used to detect CNVs. However, similar traits existed among the populations from which they were derived, and we aimed to analyze CNV differences among the common traits.

Our research highlights an interesting pattern. The number of CNVS in the WECP populations is larger, and the segments are longer than those in the AHIP populations. We considered three possible reasons to explain these observations: First, the WECP population is subjected to high intensity hybrid breeding, while the AHIP population is currently small owing to the impact of pig plague in Africa, leading to the possible existence of some inbreeding. These factors may result in a higher genetic polymorphism in WECPs than in AHIPs, as reports have found that some CNVs may segregate in inbred populations [31]. The second possibility is because the tool we employed was the Porcine 80K SNP BeadChip, which has a high marker density, but the reference genome used in the analysis was based on the Duroc genome, which could have resulted in higher alignment for WECPs. Thirdly, the five breeds in the AHIP population have similar origins and may have had gene communication at an early stage or even have shared ancestry. This may not be the case with the breeds in the WECP population, which have independent origins. We compared our results with those of other studies; and found that the CNVs detected in this study were greater than those identified in the Large White (LW) population by Wang et al. [32], who used similar detection methods (Appendix A). Approximately 181.71 Mb (32.85%) in the WECP population in this study overlapped with their results; we only contrasted the loss and gain events, and both events were not counted. The reason for this discrepancy may be that the LW breed was included in our study, while there was only one with the large number in Wang et al. study. Therefore, a certain degree of similarity was observed. However, differences in domestication selection and genetic background contribute equally to the occurrence of this phenomenon.

We next annotated the genes within the CNVR regions in the WECP and AHIP populations and performed enrichment analyses separately by loss and gain event types and identified several genes associated with disease resistance in the WECPs_loss event. Forkhead Box J1 (*FOXJ1*) and Forkhead Box K2 (*FOXK2*) are among the Forkhead Box (Fox) family members [33], and numerous studies have shown that Fox family members play an important role in lymphatic system development and immune function regulation [34]. Ma et al. [35] found that the transcriptional and protein levels of the host factor FOXJ1 were significantly downregulated by African swine fever virus (ASFV) infection of primary porcine alveolar macrophages. Multiple studies confirm that FOXJ1 plays an antiviral role against ASFV replication, revealing a function for FOXJ1 in positively regulating the innate immune response. FOXK2 was previously defined as an ILF-enhancer binding factor and has now been shown to bind to the IL-2 promoter [36,37], a transcription factor essential for IL-2 mRNA synthesis in activated T lymphocytes [38]. In human studies, the disruption of *FOXK2* was found to be associated with central nervous system abnormalities and intellectual disability [39]. Using an SNP BeadChip, Fabian et al. [40] identified some CNVs in a population with anorectal malformations (ARM) disease, confirmed the presence of nine submicroscopic CNVs using qPCR and suggested several genes, such as *FOXK2*, as factors causing ARM disease. The mannose-binding lectin 2 (*MBL2*) gene is one of the members of the C-type (Ca^2+^ dependent) lectin gene superfamily and plays an important role in the first line of defense against pathogen infection [41]. The MBL-C protein encoded by the *MBL2* gene is an important component of the innate immune lectin pathway and functions to mediate immune regulation and phagocytosis [42]. Many studies have shown that *MBL2* is an important component of the innate immune response and a key gene regulating the resistance or susceptibility of livestock to pathogens [43,44]. Álvarez et al. demonstrated that *MBL2* underlies a QTL for immunoglobulin A levels on OAR22 of Djallonké lambs and confirms a signature of resistance to infection with gastrointestinal parasites [45]. In addition, we also found loss of TNF receptor superfamily member 4 (*TNFRSF4* [46]). *TNFRSF4* is one of the major members of the TNF receptor-family member 4 (TN-FRSF4) and is mainly inducibly expressed in activated CD4+ and CD8+ cells [47]. TNFRSF4, combined with its ligand, can promote the proliferation of T cells, prevent the development of immune tolerance [48], and play an important role in the immune environment [49]. In our study, we found that *TNFRSF4* was lost in WECP populations. Similarly, sirtuin 1 (*SIRT1* [50]), a nicotinamide adenine dinucleotide (NAD+)-dependent deacetylase, is a key regulator of cell metabolism [51]. *SIRT1* is involved in regulating the activation of macrophages and T lymphocytes via the NF-kB and AP-1 pathways, inhibiting inflammation as well as promoting cellular stress resilience by improving mitochondrial function [52]. In studies of mice, Labiner found that a knockout of *SIRT1* in mice exacerbated renal mitochondrial dysfunction and increased inflammation and mortality in sepsis [53]. The neutrophil cytosolic factor 1 (*NCF1* [54]) gene was also lost in the WECP population, and all of them affected the immunity of individuals in earlier studies reported. Therefore, we conjecture that the loss of these genes has the potential to be one of the important factors that contributes to the higher resistance and immunity of Chinese domestic pigs compared to foreign pig breeds.

Our team have previously reported significant differences in meat quality traits and fat deposition ability between WECP and AHIP populations [55,56,57]. This study also detected some genes related to intramuscular fat deposition in the CNV gene enrichment analysis that were loss events in the WECP population. The diacylglycerol acyltransferase 1 (*DGAT1*) gene belongs to the family of cholesterol acyltransferases (ACAT) and has been shown to significantly increase intramuscular fat content in domestic animals, such as pigs [58], cattle [59], and sheep [60]. Li et al. [61] found that the overexpression of *DGAT1* elevates intramuscular fat (IMF) content in transgenic mice generated by pronuclear microinjection with a muscle-specific promoter of porcine muscle creatine kinase (MCK). Liu et al. [62] were the first to detect the distribution of *DGAT1* CNVs in seven Chinese goat populations and found significant associations with important economic traits such as meat, dairy, and fiber goats. In a porcine study, Wang et al. [63] found some CNV genes related to lipid metabolism, such as *DGAT1* and *DGAT2*, in the Duroc population. In our study, we identified ten QTLs, including average daily gain (ID = 18712441), average backfat thickness (ID = 8134840), water holding capacity (ID = 15473302), and linoleic acid content (ID = 17965327) by gene mapping to the CNVs in the WECPs_loss group and by using functional analysis of the location of the CNVR in *DGAT1* using the PigQTLdatabase version 10.2. Domestic animals undergo a series of changes in meat quality over a period of time after being slaughtered. Among them, the concentration of inosine 5′-monophosphate (IMP) has been suggested to be an important factor affecting meat quality [64] because IMP is one of the major nucleotides present in the post-mortem muscle of domestic animals produced by the rapid degradation of ATP to ADP and AMP, followed by further degradation to IMP [65,66]. Related studies have reported that IMP combined with glutamic acid or aspartic acid enhances the umami taste in meat [67]. The ecto-5′-nucleotidase (*NT5E*) gene encodes the enzyme NT5E for the extracellular degradation of IMP to inosine, and several studies have simultaneously reported that NT5E affects the overall concentration balance of IMPs and their degradation products [68]. Komatsu et al. [69] estimated the genetic parameters of IMP and its degradation products at four and seven days post-mortem in cattle by using a model with or without the *NT5E* genotype. The results showed that the heritability of IMP at seven days post-mortem decreased from 0.32 to 0.08 when the *NT5E* genotype was included in the model. Similarly, *NT5E*, which is involved in the biological process of lipid deposition, has been identified as a candidate gene for selecting markers to control IMF formation in Chaohu ducks [70]. In this study, we identified 22 QTLs, including those related to meat quality, such as marbling (ID = 38049), muscle movement percentage (ID = 38073), and drip loss (ID = 21947,), based on CNVs located in the NT5E gene and the QTL database in control pigs. 

Although we found some interesting information in WECP and AHIP populations, our study did have some limitations. We used the Porcine 80K SNP BeadChip, which has limitations that cannot be avoided by showing the specific locations of all CNVs somatically. Fortunately, the beadchip employed in our study was large enough for its loci to be mapped relative to other beadchips on pigs. However, the number of groups we studied was not very large, owing to the presence of African pig fever, which affected the number of our groups. Therefore, in future experiments, we plan to expand the study population and obtain phenotypic data to validate the effect of CNVs.

## 5. Conclusions

In this study, we evaluated CNV differences between the WECP and AHIP populations, based on the Porcine 80K SNP BeadChip. We found 225 and 379 CNVRs in the WECP and AHIP populations, respectively, accounting for 22.57% and 10.90% of the porcine autosomal genome. Statistically, the number of CNVRs was greater and their length was longer in the WECP than in the AHIP population. We performed enrichment analysis separately by CNVR event type and found several immunity and meat quality-related candidate genes in the WECP-loss group. Our analyses found that these genes overlapped with important economic traits in pigs. These results can help us understand the reasons for the differences in phenotype between the WECP and AHIP populations and help to effectively improve the performance of these breeds.

## Figures and Tables

**Figure 1 genes-14-00654-f001:**
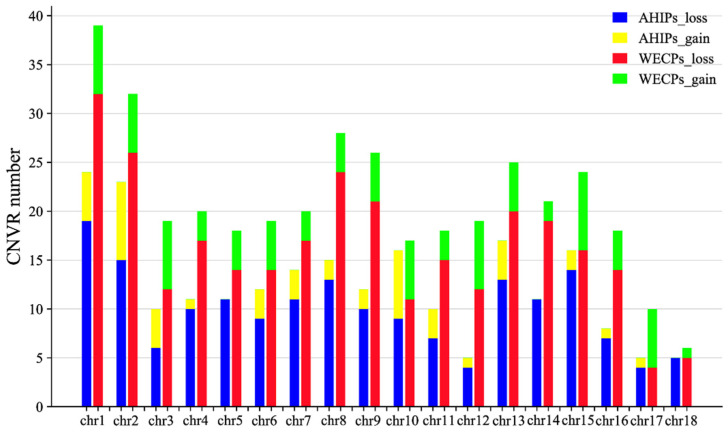
Statistical bar graphs of CNVRs in autosomes of the AHIP and WECP populations. A total of 604 CNVRs in the AHIP and WECP populations were on autosomes, colored in blue (AHIPs_loss), yellow (AHIPs_gain), red (WECPs_loss), and green (WECPs_gain), respectively. *Y*-axis values are the number of CNVRs and *X*-axis values are chromosomes.

**Figure 2 genes-14-00654-f002:**
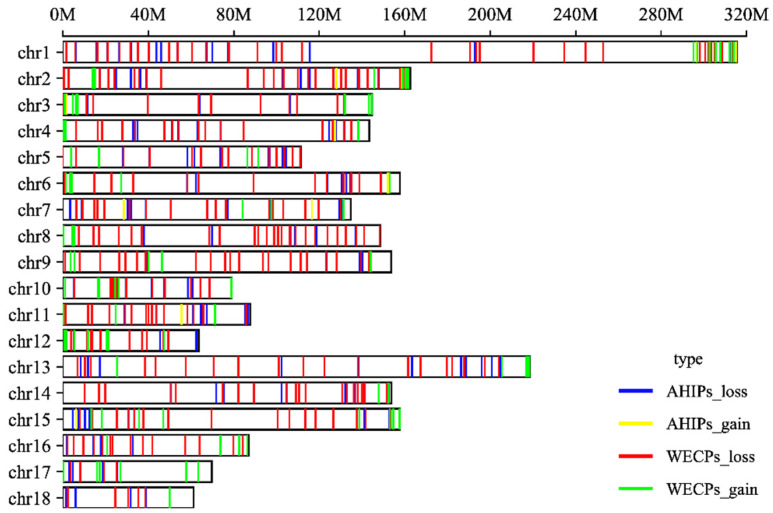
Genomic distribution and status of CNVRs in pig autosomes. The locations of 604 CNVRs are shown on the autosomes in blue (AHIPs_loss), yellow (AHIPs_gain), red (WECPs_loss), and green (WECPs_gain). Y-axis values are chromosome names, and X-axis values are chromosome position in Mb, which are based on the Sscrofa 10.2 reference genome assembly.

**Figure 3 genes-14-00654-f003:**
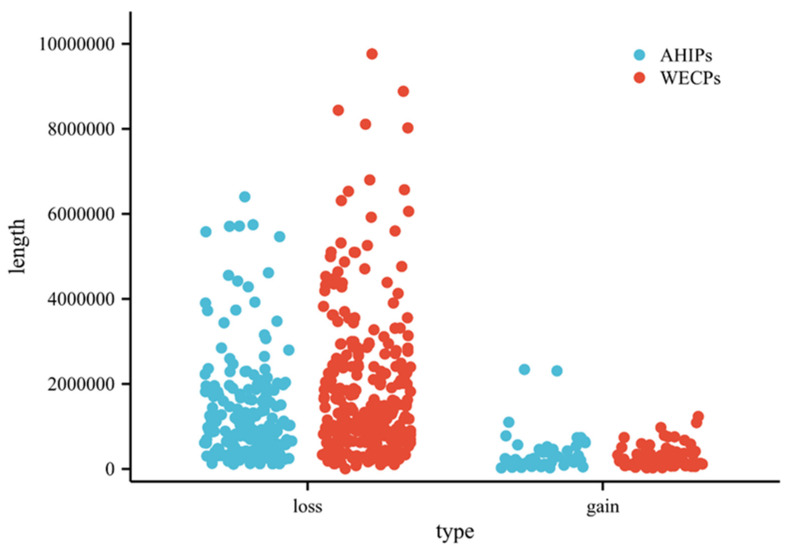
Different CNVR events and length statistics in the AHIP and WECP populations. The *X*-axis represents the type of CNVR in the AHIP and WECP populations, and the *Y*-axis is the length size of the CNVR. Violin width is wider representing more data and narrower representing less data.

**Figure 4 genes-14-00654-f004:**
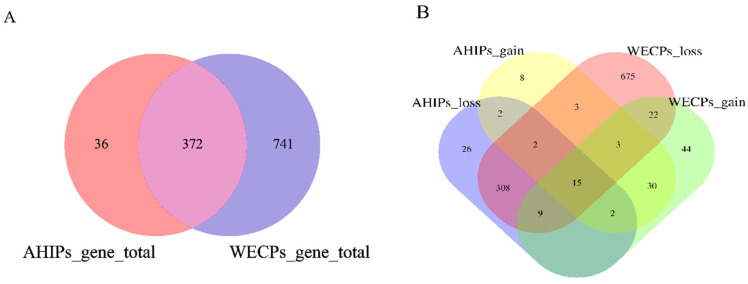
Venn diagram of CNVRs and related genes in AHIP and WECP populations. (**A**) Venn diagram of annotated total genes in AHIP and WECP populations. (**B**) Venn diagram of different CNVR event annotated total genes in AHIP and WECP populations.

**Figure 5 genes-14-00654-f005:**
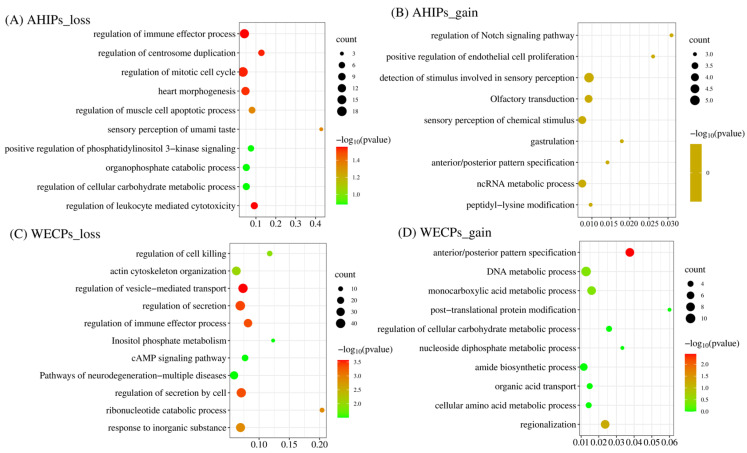
GO and KEGG pathways analysis of the annotated genes in CNVR in the AHIP and WECP populations, and the green to red indicate the multi-test adjusted p-value in log base 10. (**A**) The GO and KEGG pathway analysis results of annotation genes in CNVR of AHIPs_loss event. (**B**) The GO and KEGG pathway analysis results of annotation genes in CNVR of AHIPs_gain event. (**C**) The GO and KEGG pathway analysis results of annotation genes in CNVR of WECPs_loss event. (**D**) The GO and KEGG pathway analysis results of annotation genes in CNVR of WECPs_gain event.

**Table 1 genes-14-00654-t001:** The distribution of CNVRs in the pig autosomes.

Group	Sample Size	Total Number of CNV	Unique Number of CNVR	Gain	Loss	Average Size (kb)	Max Size (kb)	Median Size (kb)	Min Size (kb)
AHIPs	170	3863	225	47	178	1187.45	6400.56	774.74	25.44
WECPs	150	7546	379	86	293	1459.32	9761.85	901.97	4.85
Total	320	11409	604	133	471	1358.04	9761.85		4.85

## Data Availability

The datasets presented in this study can be found in online repositories. The names of the repository/repositories and accession number(s) can be found in the article. The dataset used and analyzed during the current study is available from the corresponding author on reasonable request.

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
