# Peer review of "Genome-Wide Detection and Analysis of Copy Number Variation in Anhui Indigenous and Western Commercial Pig Breeds Using Porcine 80K SNP BeadChip"

_genes, 2023, doi:10.3390/genes14030654_

Round 1

Reviewer 1 Report

The manuscript (MS) describes copy number variation (CNV) in Anhui indigenous and Western commercial pig breeds analysed via SNP-chip. The study utilises enough animals and highlights regions of CNVs where gains and losses occurred. the MS also gives a list of genes within the above-mentioned regions associated with quality traits. 

Results are discussed in view of other results found in the literature.  Plausible explanations are given for the "interesting pattern" (Page 8, line 198) described by the authors. I think the second explanation "...the reference genome used in the analysis was based on the Duroc genome, this could have resulted in higher alignment for commodity pigs." could be checked by looking at the experimental data. Maybe the result will just further validate the "interesting pattern".

There may be more, but I have found two acronyms not defined in the MS:  LRR and BAF. Please define them and check other acronyms too.

Author Response

Dear Editors and Reviewers:

Thank you for your letter and for the reviewers’ comments concerning our manuscript entitled “Genome-wide detection of Copy number variation analysis in Anhui indigenous and Western commercial pig breeds using PorcineSNP80k data” (ID: genes-2200248). Those comments are all valuable and very helpful for revising and improving our paper, as well as the important guiding significance to our researches. We have studied comments carefully and have made correction which we hope meet with approval. Revised portion are marked in red in the paper. The main corrections in the paper and the responds to the reviewer’s comments are as flowing:

Comments and Suggestions for Authors

The manuscript (MS) describes copy number variation (CNV) in Anhui indigenous and Western commercial pig breeds analysed via SNP-chip. The study utilises enough animals and highlights regions of CNVs where gains and losses occurred. the MS also gives a list of genes within the above-mentioned regions associated with quality traits. 

Results are discussed in view of other results found in the literature.  Plausible explanations are given for the "interesting pattern" (Page 8, line 198) described by the authors. I think the second explanation "...the reference genome used in the analysis was based on the Duroc genome, this could have resulted in higher alignment for commodity pigs." could be checked by looking at the experimental data. Maybe the result will just further validate the "interesting pattern".

Response: Thank you very much for your suggestions. Your suggestions are helpful to our modification, and we have modified them. Line 236-248

There may be more, but I have found two acronyms not defined in the MS:  LRR and BAF. Please define them and check other acronyms too.

Response: We are very sorry that we have modified this due to our mistakes. Line 107-108

Reviewer 2 Report

Review report for the manuscript “Genome-wide detection of Copy number variation analysis in Anhui indigenous and Western commercial pig breeds using PorcineSNP80k data”

Summary: aim, main contributions and strengths of manuscript

Aim: Characterization of Copy Number Variation in the genomes of indigenous Chinese and commercial pig breeds.

Main contributions: The study increases the depth of our understanding the role of structural variation in the performance of indigenous Chinese and commercial pig breeds. Further, the study suggests candidate genes, for example, FOXJ1 (within the Copy Number Variation Regions) involved in functions related to health, growth and meat quality in the populations under study.

Strengths: The genomic background of the study population supports the analysis and inferences.

General concept comments

Article

Abstract: The abstract  has a word count of 253. The aim of the study is not clearly stated (refer to guidelines for authors). The abbreviations AHIPs and WECPs should be clearly defined at the point of their first use. Most of the abstract should be re-written to clearly communicate what the authors intend to state.

Introduction: Sections should be re-written to clearly communicate what the authors intend to state.

Materials and methods: More detail on the selection/ inclusion criteria for the pig breeds would enrich the manuscript. Were these the same populations used in the previous study and hence the choice of groups? It is not clear if breed purity is a concern in this study given the history of pig domestication. This is important since genomics is pivotal to this study. It would be of value if the authors highlighted their preference for the Sus scrofa genome build 10.2 for most of their work and only referring to build 11.1 (a more recent version) in the latter sections. While the manuscript’s title implies genome-wide sequencing, the authors actually base their work on the Porcine 80K SNP Chip (SNP array) hence some of the shortcomings faced in the analysis. Sections of the methodology should be rewritten to clearly state what was done for reproducibility.

Review

The review is supported by 61 appropriately referenced publications. The major part of the review is dedicated to information about the CNVs which could be reduced. Details of what is known about CNVs in indigenous Chinese and Commercial pig populations would further highlight the gap the study seeks to close and support the hypothesis advanced by the authors. Considering that the previous study forms the motivation for the current one, it would be of value to concisely state that basis. Sections of the literature review should be rewritten for clarity.

Specific comments

Abstract

Lines 16-17: clearly indicate that there were two populations (AHIP and WECP)

Lines 17-21 Split into two concise sentences.

Lines 21-22 consider rephrasing e.g. “We referred to the porcine genome to annotate..”

Lines 23-24: which traits? and what does missing status mean?

Lines 24-25: please indicate if these were detected in AHIP or WECP.

Lines 25-27: This sentence seems misplaced because it restates results presented in the previous ones. Further, which differences in immunity and meat quality do the authors refer to?

Lines 27-29: Was performance evaluation conducted in this study?

Introduction

Line 34: replace “kind” with “kinds”

Lines 39-41: check grammar

Lines 41-43: check grammar

Line 45: “Pig” is not a genus. Please consider Sus? Sus domesticus?

Lines 53-57: rephrase into one sentence

Line 62: check grammar viz inherited by / passed to offspring

Materials and methods

Line 94: Citation?

Line 95: These protocols are several. Please specify which one.

Line 104: A SNP Chip contains autosome SNPs and those found on the sex chromosomes. It is expected that this value would be lower than 868,528 after QC. Please confirm this result.

Line 111: Please ensure that the website address is correct

Line 122: check spelling "m1anual"

Line 122: is this the correct in text citation style? (Kumah et al)

Lines 131-134. Are these two separate sentences? What was Metascape used for?

Line 139: confirm that this is the correct website address

Results

No line number: Figure 1- Is this a bar chart/ graph? What does "statistical mapping" mean?

No line number: Figure3- These look like violin plots. Punctuate legend/ caption correctly.

Line 166: check spelling "Scrofa"

Line 166: “studies” or “analyzes”?

No line number: Figure 5- can be improved for reading ease

Discussion

Line 189: “impact” or “effect”?

Line 190: “structural variation” or “genetic structural variation”?

Line 191: “qualities” or “traits” or “characteristics”? All traits are phenotypic with some production and others economically relevant (important)

Lines 193-195: should be re-written to communicate precisely what the authors intend to state.

Lines 195-196: Please provide a reference to this effect. SNP arrays have inherent limitations on resolution and sensitivity when used to detect CNVs.

Lines 198-204: The discussion should maintain the groupings (AHIP and WECP) rather than introduce new terms viz commodity pigs (or commercial pigs?). This section should be rewritten to clearly communicate what the authors intend to state. References should be provided around effects of inbreeding on occurrence of CNVs/ CNVRs. Further, what is the inbreeding rate/ level in Chinese indigenous pigs at least for the breeds included in this study? What is meant by small population considering China is a top pig producer.

Line 209: “communication” or “introgression”? reference to this event?

Line 212: replace outnumbered with "were more than"

Lines 216-217: reference to this?

Lin 221: This is an interesting finding and the authors should clearly state that these genes relate to the WECP_loss when reporting it in the abstract

Line 235: delete “f” before FOXK2

Lines 251-252 is mentioned in the abstract and implies that this study run such an analysis. However, here it comes from a previous study. The authors should report only findings of the current study in the abstract.

Line 267: PigQTLdb?

Conclusions

Lines 287-289: These sentences need to be rephrased into a point each or joined into one to clearly communicate what the authors intend to state.

Lines 289-294: This needs to be rephrased to communicate what the authors intend to state.

Institutional Review Board Statement

Line 309: Should “Animal Care Committee” be the Institute Animal Care and Use Committee (IACUC)?

Acknowledgments

Lines 316-318: It is not clear what the abbreviations mean.

References

61 and appropriately cited

Author Response

Dear Editors and Reviewers:

Thank you for your letter and for the reviewers’ comments concerning our manuscript entitled “Genome-wide detection of Copy number variation analysis in Anhui indigenous and Western commercial pig breeds using PorcineSNP80k data” (ID: genes-2200248). Those comments are all valuable and very helpful for revising and improving our paper, as well as the important guiding significance to our researches. We have studied comments carefully and have made correction which we hope meet with approval. Revised portion are marked in red in the paper. The main corrections in the paper and the responds to the reviewer’s comments are as flowing:

Comments and Suggestions for Authors

Review report for the manuscript “Genome-wide detection of Copy number variation analysis in Anhui indigenous and Western commercial pig breeds using PorcineSNP80k data”

Summary: aim, main contributions and strengths of manuscript

Aim: Characterization of Copy Number Variation in the genomes of indigenous Chinese and commercial pig breeds.

Main contributions: The study increases the depth of our understanding the role of structural variation in the performance of indigenous Chinese and commercial pig breeds. Further, the study suggests candidate genes, for example, FOXJ1 (within the Copy Number Variation Regions) involved in functions related to health, growth and meat quality in the populations under study.

Strengths: The genomic background of the study population supports the analysis and inferences.

General concept comments

Article

Abstract: The abstract has a word count of 253. The aim of the study is not clearly stated (refer to guidelines for authors). The abbreviations AHIPs and WECPs should be clearly defined at the point of their first use. Most of the abstract should be re-written to clearly communicate what the authors intend to state.

Response: Thank you for your suggestion. We have condensed the summary to ensure compliance with the requirements.

Introduction: Sections should be re-written to clearly communicate what the authors intend to state.

Response: We are very sorry for our wrong writing. We have adjusted the introduction as a whole.

Materials and methods: More detail on the selection/ inclusion criteria for the pig breeds would enrich the manuscript. Were these the same populations used in the previous study and hence the choice of groups? It is not clear if breed purity is a concern in this study given the history of pig domestication. This is important since genomics is pivotal to this study. It would be of value if the authors highlighted their preference for the Sus scrofa genome build 10.2 for most of their work and only referring to build 11.1 (a more recent version) in the latter sections. While the manuscript’s title implies genome-wide sequencing, the authors actually base their work on the Porcine 80K SNP Chip (SNP array) hence some of the shortcomings faced in the analysis. Sections of the methodology should be rewritten to clearly state what was done for reproducibility.

Review

The review is supported by 61 appropriately referenced publications. The major part of the review is dedicated to information about the CNVs which could be reduced. Details of what is known about CNVs in indigenous Chinese and Commercial pig populations would further highlight the gap the study seeks to close and support the hypothesis advanced by the authors. Considering that the previous study forms the motivation for the current one, it would be of value to concisely state that basis. Sections of the literature review should be rewritten for clarity.

Specific comments

Abstract

Lines 16-17: clearly indicate that there were two populations (AHIP and WECP)

Response: We are very sorry for our erroneous writing, for which we have revised. Line 17

Lines 17-21 Split into two concise sentences.

Response: We have made correction according to the Reviewer’s comments. Line 18-21

Lines 21-22 consider rephrasing e.g. “We referred to the porcine genome to annotate..”

Response: We refer to the requirements for the number of words in the author's guide. This part is not very important, and we decided to delete it.

Lines 23-24: which traits? and what does missing status mean?

Response: I'm sorry we didn't describe it clearly enough, “missing status” mean these genes may be absent in the WECPs population but normal in the AHIPs population.

Lines 24-25: please indicate if these were detected in AHIP or WECP.

Response: The enrichment analysis we performed in terms of CNV status in both populations, which we obtained for genes that were in deletions in the WECPs population but that were normal in the AHIPs population.

Lines 25-27: This sentence seems misplaced because it restates results presented in the previous ones. Further, which differences in immunity and meat quality do the authors refer to?

Response: As Reviewer suggested that we modified the abstract as a whole. Thank you very much for your suggestions. Line 13-25.

Lines 27-29: Was performance evaluation conducted in this study?

Response: We did not conduct a performance evaluation. This is the goal we look forward to, and we plan the next steps to validate our results.

Introduction

Line 34: replace “kind” with “kinds”

Response: Sorry very much for our negligence, we have modified. Line 32.

Lines 39-41: check grammar

Response: Thank you for your suggestion, and we have revised it. Line 41-43

Lines 41-43: check grammar

Response: Thank you for your suggestion, and we have revised it. Line 43-46

Line 45: “Pig” is not a genus. Please consider Sus? Sus domesticus?

Response: We are very sorry for the wrong way of writing. We have corrected this. Line 43

Lines 53-57: rephrase into one sentence

Response: Thank you very much for your suggestion. We have corrected it. Line 63-68.

Line 62: check grammar viz inherited by / passed to offspring

Response: Thanks for your suggestions, we revised. Line 49

Materials and methods

Line 94: Citation?

Response: Sorry very much for our negligence, here we have corrected. Line 94

Line 95: These protocols are several. Please specify which one.

Response: Herein we corrected the instruction. Line 94-95

Line 104: A SNP Chip contains autosome SNPs and those found on the sex chromosomes. It is expected that this value would be lower than 868,528 after QC. Please confirm this result.

Response: Sorry very much for our negligence, by reviewing the original data, indeed as you said, we corrected it. Line 104

Line 111: Please ensure that the website address is correct

Response: Thank you very much for your guidance on this article. Line 111-112

Line 122: check spelling "m1anual".

Response: Sorry very much for our mistake, we made correction. Line 122

Line 122: is this the correct in text citation style? (Kumah et al)

Response: Sorry very much for our miscommunication and we have made correction. Line 122.

Lines 131-134. Are these two separate sentences? What was Metascape used for?

Response: We use GCModel in PennCNV software, which combines our preparation documents to correct the influence of GC gene wave. The ultimate goal is to avoid false positives. And the Metascape is a software used for functional enrichment analysis.

Line 139: confirm that this is the correct website address

Response: Thank you very much for your guidance on this article. Line 150

Results

No line number: Figure 1- Is this a bar chart/ graph? What does "statistical mapping" mean?

Response: I'm very sorry for our wrong writing. Figure 1 is a bar chart. Line:

No line number: Figure3- These look like violin plots. Punctuate legend/ caption correctly.

Response: Figure 3 is a picture of violin

Line 166: check spelling "Scrofa"

Response: Thank you very much for your suggestion. We have corrected it. Line181

Line 166: “studies” or “analyzes”?

Response: Thank you very much for your suggestion. We have corrected it. Line180

No line number: Figure 5- can be improved for reading ease

Response: Thank you very much for your suggestion. At present, I am still considering how to optimize the picture. However, no suitable template has been found so far, but I am still working on this task due to time problems.

Discussion

Line 189: “impact” or “effect”?

Response: Sorry very much for our negligence, for which we can revise.

Line 190: “structural variation” or “genetic structural variation”?

Response: We considered that it should be structural variation, which was corrected

Line 191: “qualities” or “traits” or “characteristics”? All traits are phenotypic with some production and others economically relevant (important)

Response: Thanks for your suggestions, we have modified this. Line 221-223.

Lines 193-195: should be re-written to communicate precisely what the authors intend to state.

Response: In order to express what we stated, this was re described and if you do not understand, I try to continue with the revision in order to guarantee that the reader understands. Line 227-229

Lines 195-196: Please provide a reference to this effect. SNP arrays have inherent limitations on resolution and sensitivity when used to detect CNVs.

Response: Thank you very much for your help and indeed this factor is present as you stated, and I apologize for our consideration step and also thank you for your suggestion. Line 229-231

Lines 198-204: The discussion should maintain the groupings (AHIP and WECP) rather than introduce new terms viz commodity pigs (or commercial pigs?). This section should be rewritten to clearly communicate what the authors intend to state. References should be provided around effects of inbreeding on occurrence of CNVs/ CNVRs. Further, what is the inbreeding rate/ level in Chinese indigenous pigs at least for the breeds included in this study? What is meant by small population considering China is a top pig producer.

Response: Thank you for your suggestions. Previously we were able to find, by reading the literature, that high levels of near intersection contribute to CNVs in cohorts of mice and chickens. This is true in the present study where the number of AHIPs populations declined due to the effects of pig plague in Africa, but we cannot determine how much inbreeding coefficient can be achieved in the AHIPs population. But in a previous study we found that there was gene communication in the AHIPs population. For the context of the population we have investigated, there is a certain degree of inbreeding. We modified the description and interpretation in the article. Line:236-241

Line 209: “communication” or “introgression”? reference to this event?

Response: We corrected for this. Line:245-248

Line 212: replace outnumbered with "were more than"

Response: We have revised this and thank you for your suggestions. Line:249.

Lines 216-217: reference to this?

Response: Sorry very much for our wrong writing, which we describe more clearly in the article. Line: 253-257.

Lin 221: This is an interesting finding and the authors should clearly state that these genes relate to the WECP_loss when reporting it in the abstract

Response: Thank you., your suggestion is exactly what we want to express in the summary. So our error description method needs to be improved, and we have corrected it.

Line 235: delete “f” before FOXK2

Response: I'm sorry for my negligence. We have corrected it.

Lines 251-252 is mentioned in the abstract and implies that this study run such an analysis. However, here it comes from a previous study. The authors should report only findings of the current study in the abstract.

Response: Sorry very much for our description of problems. Our previous findings were not in our present study, and this part is just a summary of our previous studies. And the latter results were in accordance with the enrichment analysis we screened, so these genes were not the result of the previous study.

Line 267: PigQTLdb?

Response: We are very sorry for our error description. We can correct it.

Conclusions

Lines 287-289: These sentences need to be rephrased into a point each or joined into one to clearly communicate what the authors intend to state.

Response: Sorry very much for the problems with our writing and we have revised the whole Conclusions section.

Lines 289-294: This needs to be rephrased to communicate what the authors intend to state.

Response: Sorry very much for the problems with our writing and we have revised the whole Conclusions section.

Institutional Review Board Statement

Line 309: Should “Animal Care Committee” be the Institute Animal Care and Use Committee (IACUC)?

Response: Very sorry, we verified with articles published in previous studies and found no problems

Acknowledgments

Lines 316-318: It is not clear what the abbreviations mean.

Response: We refer to the Microsoft version guide for description, with capital letters representing initials of names.

References

61 and appropriately cited

Reviewer 3 Report

The paper is well-written and complete. The content matches the title. The methods and interpretation of the data are valid. I have only one comment for the authors: I suggest including in the discussion section some sentences about the practical implications (how could be used in the context of a breeding program and what is their impact on genetic evaluations) of this analysis.

Author Response

Dear Editors and Reviewers:

Thank you for your letter and for the reviewers’ comments concerning our manuscript entitled “Genome-wide detection of Copy number variation analysis in Anhui indigenous and Western commercial pig breeds using PorcineSNP80k data” (ID: genes-2200248). Those comments are all valuable and very helpful for revising and improving our paper, as well as the important guiding significance to our researches. We have studied comments carefully and have made correction which we hope meet with approval. Revised portion are marked in red in the paper. The main corrections in the paper and the responds to the reviewer’s comments are as flowing:

Response: Thank you very much for your suggestion. We have made extensive rectification in the initial revision and supplemented it according to your opinion.

Round 2

Reviewer 2 Report

The manuscript has greatly improved in terms of its content. However, English language and style should be improved further for the benefit of the reader.

Author Response

Dear Reviewers:

Thank you for your letter and for the reviewers’ comments concerning our manuscript entitled “Genome-wide detection of Copy number variation analysis in Anhui indigenous and Western commercial pig breeds using PorcineSNP80k data” (ID: genes-2200248). Those comments are all valuable and very helpful for revising and improving our paper, as well as the important guiding significance to our researches. We have studied comments carefully and have the guidance of professionals which we hope meet with approval. Revised portion are marked in red in the paper.